# Prediction of the Net Energy of Wheat from Chemical Analysis for Growing Ducks

**DOI:** 10.3390/ani13061097

**Published:** 2023-03-20

**Authors:** Yanru Liang, Qinteng Hou, Mengchao Yu, Yaqi Chang, Hua Zhao, Guangmang Liu, Xiaoling Chen, Gang Tian, Jingyi Cai, Gang Jia

**Affiliations:** 1Institute of Animal Nutrition, Key Laboratory of Animal Disease-Resistance Nutrition Ministry of Education, Sichuan Agricultural University, Chengdu 611130, China; 2Key Laboratory of Animal Disease-Resistant Nutrition and Feed, Ministry of Agriculture and Rural Affairs, Chengdu 611130, China

**Keywords:** duck, net energy requirement, regression method, comparative slaughtering method, chemical composition, prediction equation

## Abstract

**Simple Summary:**

For this experiment, the Cherry Valley duck was selected as the experimental animal. By measuring the chemical composition of wheat from different sources and net energy values, we analyzed the correlation between the NE and the chemical composition of wheat. Finally, the best-fit equation was established: NE = 0.380 AME − 0.147 NDF − 0.274 ADF + 5.262 (R^2^ = 0.874, RSD = 0.19, *p* < 0.001).

**Abstract:**

The goal of this study was to determine the net energy (NE) value of wheat for growing ducks and establish a NE prediction equation based on the grain’s chemical composition. Forty wheat samples were selected based on bulk weight from major wheat-producing regions in China. A total of 460 1-week-old ducks (initial body weight (BW): 134.86 ± 3.32 g) were randomly assigned to 46 diets, including a basal diet, 5 restricted feeding diets and 40 test diets. Each diet contained five replicates, each with two ducks. The basic diet was a corn–soybean meal, and 40 kinds of experimental diets were prepared by mixing the basic diet with 20% wheat. A prediction equation for the NE concentration was created using the chemical make-up of wheat samples. The results indicated that the NE and apparent metabolism energy (AME) content of 40 wheat samples ranged from 6.81 to 9.12 MJ/kg and from 11.03 to 14.34 MJ/kg, respectively. The ether extract (EE), neutral detergent fiber (NDF), acid detergent fiber (ADF) and AME were highly correlated with NE value (*p* < 0.01), with the AME and NE showing the strongest correlation (r = 0.884). Chemical features could be used to predict the NE values with accuracy, and the prediction equation was strengthened by the inclusion of the AME. The best-fit equation was as follows: NE = 0.380 AME − 0.147 NDF − 0.274 ADF + 5.262 (R^2^ = 0.874, RSD = 0.19, *p* < 0.001). In summary, the NE value of wheat is 8.49 ± 0.30 MJ/kg for growing ducks, and the chemical composition can be used to accurately predict NE in wheat.

## 1. Introduction

With the improvement of poultry breeding level and the progress of feed processing technology, the feed industry has developed rapidly, and the demand for feed raw materials is also increasing. With the transformation of meat duck breeding systems from traditional free range to large-scale and intensive breeding modes, the demand for feed has increased significantly.

Currently, the preparation of a poultry diet is basically based on metabolic energy (ME). In contrast to the ME system, NE takes into account the total energy loss of the feed in the animal’s energy utilization process, as well as the differences in metabolic efficiency of different nutrients, and is thus gradually favored by researchers [1,2]. It has been widely used in the pig industry, but less research has been conducted on NE in poultry than in pigs. Although the chemical analysis prediction equation for evaluating the NE of meat duck feed raw material has been studied, the establishment of the prediction equation for the NE of specific feed raw material or the prediction equation containing different feed NE still needs further research [3].

Wheat is a major crop in China and is widely used in poultry feed as an energy source. However, the energy value of wheat from different sources is varied for its chemical composition and nutrient level, which can affect the economic value and dietary inclusion rate in feed formulations [4]. Therefore, it is critically important to estimate the accurate energy values of different wheats. However, ME systems can overestimate the energy value of fiber and high-protein feedstuffs, while underestimating the feedstuffs that have high concentrations of starch and fat [5]. This has made it difficult to utilize wheat resources reasonably and effectively. Therefore, if an NE system can be used to evaluate the energy value of wheat, establish the chemical prediction equation for rapid determination of the energy value of wheat, and accurately configure the poultry wheat diet, it will be able to accurately evaluate the effective utilization rate of wheat for meat ducks, promote the application of a NE system in meat duck feed, and be of great significance for the high-efficiency use of meat duck feed. However, there is no report on the NE value of meat duck made of wheat.

The present study was undertaken to determine the NE content of wheat using comparative slaughter in growing ducks and to establish a NE prediction equation based on the chemical composition of wheat to provide an experimental basis for the construction and application of a NE system.

## 2. Materials and Methods

The experimental procedures for animal trials were conducted in accordance with the Chinese guidelines for animal welfare and were approved by the Animal Health and Care Committee of Sichuan Agricultural University (No. 20180718).

### 2.1. Wheat Sample Collection

To increase the accuracy of the prediction equations, 40 kinds of wheat samples were collected according to the wheat production area and wheat bulk density in China, including 8 kinds in Shandong, 6 kinds in Henan and Hebei, 4 kinds in Jiangsu, 3 kinds in Anhui and Shaanxi, 2 kinds in Hubei, Shanxi and Sichuan, and 1 kind in Ningbo, Ningxia, Gansu and Yunnan.

### 2.2. Experimental Design

The experiment was evaluated by factorial method, and the NE of wheat substitute diet was divided into net energy maintenance (NEm) and net energy performance (Nep), which were determined by regression analyses and comparative slaughter technique, respectively; then, the NE value of wheat was calculated according to the set algorithm.

A total of 800 Cherry Valley ducks were selected at hatch and fed a basal diet for 5 days and 12 h. After 36 h of fasting, 480 birds with an average initial weight of 134.86 ± 3.32 g were chosen. Twenty birds were slaughtered at the beginning of the experiment to determine the initial carcass energy. NEm of 50 meat ducks was determined, and five treatment groups, including free feeding, limited feeding of 40%, 55%, 70% and 85%, were set up, with five replicates in each treatment group, and two meat ducks in each replicate were fed with basal diet. The remaining 410 meat ducks were tested for NE. The basal group and different wheat substitution groups were set up. Each treatment group had five replicates, and each replicate had two meat ducks. The meat ducks were randomly assigned to each treatment group. The basal group was fed with basal diet, and the wheat substitution group was fed with test diet. The formal period of the experiment was seven days, and the experiment began at the age of 8 days. All experimental meat ducks were fasted at 12 days and 12 h of feeding. After fasting for 36 h, all of the treated meat ducks were euthanized from cervical dislocation. The carcasses were weighed and frozen for testing.

### 2.3. Experimental Diets

A corn–soybean basal diet based on the National Research Council (1994) [6] was formulated to prepare the basic diet and the wheat test diet (Table 1).

### 2.4. Management

The experiment was conducted in the scientific research base of Sichuan Agricultural University. Before the test, the meat duck house was washed several times with a high-pressure water gun, and after it was naturally dried overnight, the shed was sprayed with benzalkonium bromide for at least 3 days, and the doors and windows were opened for ventilation for 2 days. At the same time, the tray and bucket were disinfected with benzalkonium bromide solution. The experimental ducks were caged in a line, fed and drank freely. The room temperature was maintained at 30–33 °C for the first week, then decreased by 2 °C every week, and reached room temperature level after 14 days. The relative humidity was controlled at 65–75%. The whole test area was naturally ventilated and exposed to 24 h of light.

### 2.5. Sample Collection and Processing


(1)Feed sample


After the feed was prepared, samples were taken by quartering method, each sample was about 200 g, and was sealed and stored at −20 °C to be tested. After the experiment, the feed samples were crushed and passed through a 40-mesh sieve to determine the energy value.
(2)Excretal sample

The excreta of experimental meat ducks were collected twice per day by the method of total feces collection. Then, 5% hydrochloric acid was added to the collected excreta samples for nitrogen fixation, and the samples were sealed and stored at −20 °C for testing. At the end of the collection period, all the excreta samples were put into the oven for repeated drying at 65 °C to constant weight and then humidified to make air-dried samples. After weighing, they were crushed and passed through a 40-mesh sieve, and their energy values were determined.
(3)Meat duck carcass

At the end of the experiment, the ducks were euthanized from cervical dislocation, frozen in a freezer at −20 °C, cut into pieces, and crushed. Then, the meat samples were evenly stirred and sampled, put into a freeze dryer for vacuum freeze-drying at −50 °C, and finally taken out to make air-dried samples for detection.

### 2.6. Chemical Analysis

All wheat samples were ground through a 1 mm screen for further chemical analysis. The moisture, ash, crude protein (CP), crude fiber (CF) and EE were analyzed according to the standard procedures of AOAC (2000). The CP was determined using a kjeldahl apparatus (BUCHI K-360, SW). The contents of CF, ADF and NDF were measured according to the methods of Van Soest and Wine [7] with fiber analysis equipment (Fibertec^TM^ 2010, DEN). The gross energies (GE) of wheat samples, excreta, diets and meat samples were determined using the automatic adiabatic oxygen bomb calorimeter (Parr 6400 calorimeter, Moline, IL, USA).

### 2.7. Measurements

The energy values of feed samples, excreta samples and meat samples were determined by oxygen bomb calorimeter (Parr 6400). The NE was calculated based on the equation described by Noblet: NE = NEp + NEm [8]. The NE value of wheat was calculated using the difference method by subtracting the NE contributed by the basal diet from the total NE of the diet containing a particular wheat source, the retained energy (RE) of the body was determined by the difference between the initial and final carcass’ energy content. The HP was defined as the difference between MEI and RE [9]. Regression analysis was carried out according to the logarithmic model: (Ln (HP) = Ln (a) + b × MEI) [10], and the values of a and b were obtained. When MEI is 0, the value of HP is FHP, and the meat duck NEm is calculated.

### 2.8. Statistical Analysis

Data were analyzed statistically using SPSS 22.0 (IBM, Armonk, NY, USA). The MEI, RE and HP were subjected to one-way ANOVA analysis. The linear regression equations predicted from the wheat chemical composition were performed using stepwise regression analysis. Results are presented as means and standard deviation (SD), the equations were shown with the smallest residual SD (RSD). The statistical significance level for the difference was set at *p* < 0.05.

## 3. Results

### 3.1. Physical and Chemical Characteristics of Wheats

The chemical composition and physical characteristics of the wheat samples (40 samples) are summarized in Table 2. Although little variation occurred in the CP, as well as moisture and bulk weight, a great range for some other values was observed, such as CF, ADF, NDF, EE, Ash. The bulk weight of wheat ranged from 660.4 to 863 g/L with a mean of 775.03 g/L. On a dry matter basis, the concentration of CP ranged from 10.66 to 16.16% with a mean of 14.46%. Concentrations of Ash and EE varied greatly, ranging from 1.31 to 3.29% and from 1.31 to 3.29%, respectively. The variation was also high within fiber fractions, as the NDF concentration in wheat ranged from 6.8 to 12.15% of DM (CV = 11.54%), while values for ADF in wheat ranged from 1.69% to 3.92% of DM (CV = 15.54%), suggesting a large range in the fiber content of wheat.

### 3.2. Energy Concentration, NE/AME, NEm and NEp of 40 Wheat Samples

For the 40 wheat samples, energy concentration and NE/AME are shown in Table 3. The values of NE ranged from 6.81 to 9.12 MJ/kg with a mean of 8.15 MJ/kg. The concentration of AME ranged from 11.84 to 14.34 MJ/kg (mean = 13.15 MJ/kg). The conversion rate of AME to NE ranged from 0.58 to 0.65, with an average of 0.62. In contrast, the GE content of the wheat samples varied slightly. NE requirements for duck can be partitioned into requirements for NEm and requirements for NEp. The relationship between LnHP and MEI is shown in Table 4 and Figure 1. Through the linear regression equations for lnHP as a function of MEI for duck (*p* < 0.01) and considering metabolic weight (MJ/kg BW^0.75^d^−1^), the NEm was determined for the ducks for the period 8–14 d of age:$$\mathrm{LnHP}=0.4719\mathrm{MEI}-0.570\;(R^{2}=0.9951,\;\;p<0.01)$$

In describing this relationship, a logarithmic equation was used, since extrapolation to zero MEI results in a more realistic estimate of FHP, and the calculated NE for maintenance was 566 MJ/kg BW^0.75^d^−1^. The concentrations of NEm and NEp varied greatly among these samples, ranging from 2.01 to 4.12 MJ/kg and from 3.49 to 7.36 MJ/kg, respectively.

### 3.3. Correlation among Chemical Characteristics and AME, Prediction Equations of NE

CF, NDF, and ADF were highly but negatively correlated with NE content, and the greatest correlation coefficient was observed between NDF and NE. There was a positive correlation between EE and NE content. However, no significant correlations between CP, bulk weight, and ash with NE content were found. Meanwhile, the correlation of AME with chemical characteristics was similar to NE. Correlation analysis showed that AME content of wheat has the highest correlation than any of these chemical characteristics with NE content (Table 5).

NDF, ADF, and AME were discovered to be beneficial for the NE prediction models according to the stepwise regression analysis. Equations developed from multi-step regression analysis showed that NDF was the best single predictor for NE content among chemical characteristics (R^2^ = 0.698, RSD = 0.29) (Table 6). The accuracy of prediction equations could be improved by adding ADF rather than NDF alone, with the R^2^ increased to 0.757, whereas with RSD decreased to 0.26. The results also showed that the NE content of wheat could be predicted with a reasonable degree of accuracy by measuring the AME (Equation (3) in Table 6). The addition of NDF and ADF content to the equation improved the precision of the prediction (Equations (4) and (5) in Table 6). Among the different predictors, the predictions with the highest R^2^ and lowest RSD were obtained when AME, NDF, and ADF were considered (Equation (5) in Table 6).

## 4. Discussion

### 4.1. Variation of Chemical Compositions in Wheat

The chemical composition of the wheat used in the present trials varied to a great extent. This is in agreement with previous literature data [11,12]. Kim et al. [13] reported that the CP content ranged from 9.8 to 19.1% with a mean value of 13.4%, NDF and ADF content ranged from 12.99 to 18.93% and from 2.99 to 4.42% (dry matter basis). The variation was wide for CF and ADF content, and the CP content varied widely, ranging from 12.4% to 17.4% (dry matter basis) among different wheat classes [14]. Zijlstra et al. [15] also reported that CP and CF concentrations ranged from 13.0 to 18.1% and from 2.6 to 4.1%, while NDF and ADF concentrations ranged from 12.9 to 25.0% and 3.1 to 5.1%, respectively (dry matter basis). Comparing the present mean results with the National Research Council [6] and Chinese Feed Database [16], the mean contents of CP (mean = 14.46%) and CF (mean = 3.07%) were higher than the two standards, while in NRC, the hard red winter and soft white winter means of CP were 14.15% and 11.5 %, CF = 3.0% and in CFD, means CP and CF of wheat were 13.45% and 1.9% (dry matter basis), respectively. The means of EE, Ash, NDF and ADF concentrations in this study were lower compared to the two databases. In CFD, the mean content of NDF and ADF were higher than NRC (1994), and the others were lower than the NRC (1994) values. Differences may be concerned with the analysis procedure and previous amylolytic treatment. The other reasons affecting the chemical composition of wheats are mainly due to differences in wheat cultivars, variety, growing conditions, region or season [17,18]. The previous study demonstrated that chemical composition of wheat, especially the non-starch polysaccharides (NSP) composition and structure, was significantly different due to variety and growing location and was negatively correlated with the DE content of wheat [18].

### 4.2. Determined Methods of NE, NEm and NE among Wheat

Dietary energy intake by ducks is utilized for maintenance or retention of protein or lipids, in which the energy requirement for maintenance accounts for approximately one third of total dietary energy utilization [3]. Therefore, the estimate method of NEm will influence the absolute NE value of a feed ingredient. FHP can be measured directly in fasting animals or calculated by extrapolating HP measured at different feeding levels to zero MEI, but the results of the two methods were different [19]. The use of regression to determine FHP resulted in lower FHP than the fasting method using indirect calorimetry techniques, but no difference observed in the NE of corn was determined directly or by difference reports [20]. Practically, many factors can contribute to variations in FHP, such as the length of fasting time, previous feeding level, body composition and differences in physical activity between fasted and fed animals [19,21,22,23]. Considering these factors, an alternative method for estimating the NEm is to use an exponential regression between HP and a wide range of ME intakes, which can be more accurate than a linear regression from a range of ME intake that is only above the maintenance [23], according to methodology presented by Lofgreen and Garrett [10]. In this study, we used the sample methodology such that the intercept of linear regression between the log of HP and ME intake was used to calculate NEm.

According to the comparative slaughter technique and regression analyses, Sakomura et al. [24] obtained NEm requirements that were 499.15, 376.23, and 402.81 KJ/Kg BW^0.75^d^−1^ for chickens reared at 13, 23, and 32 °C, respectively. Liu et al. [25] determined the NEm for broiler chickens to be between 386 and 462 KJ/Kg BW^0.75^d^−1^ according to different methods and regression equations. The NEm values for ducks in the current study were higher than those found in previous studies on broiler chickens. Since the NEm was associated with metabolic BW, the heavier weight of ducks may be to blame [23]. Yang et al. [26] showed that the NEm value for 2- to 3-week-old Cherry Valley ducks was 549.54 KJ/Kg BW^0.75^d^−1^ [4], which was similar to the results of this experiment. At present, different animal energy balance measurement standards are inconsistent, and the measured FHP is also different. At the same time, animal type, age, environment, body composition and other factors will affect the size of FHP. In summary, it is reliable to estimate the FHP of meat ducks by comparative slaughter method combined with the regression method.

The direct method and the substitution method are usually used for the determination of DE and ME in feed raw materials. These two methods can also be used for the determination of NE of feed raw materials. Compared with direct feeding, the substitution method has less stress on animals, and the digestion and utilization of feed and feed raw materials to be tested is more stable. In this study, the average NE value in wheat of meat ducks obtained by the substitution method is 8.15 ± 0.52 MJ/kg, and the energy efficiency of AME to NE is 0.62. Wang et al. [27] found that the NE values of soybean meal were 2.82, 6.25, 4.22 and 3.92 MJ/kg under the substitution ratios of 10%, 20%, 30% and 40%, respectively. Liu et al. [20] determined that the NE value of wheat in growing pigs by respiratory calorimetry combined with the substitution method was 11.44 MJ/kg, and the energy efficiency of AME to NE was 0.75, which was higher than that of wheat in this experiment and lower than the net energy value of 12.13 MJ/kg of growing pig wheat determined by Noblet and Shi et al. [28], and the energy efficiency of AME to NE was 0.77. This may be related to the differences in anatomical physiology and feed digestibility between meat ducks and pigs. Although the NE value of poultry feed raw materials has been reported, there is no study on NE in wheat for meat ducks.

### 4.3. Establishment of Prediction Equations of NE from Chemical Compositions

Energy is not a nutrient per se, but a quality associated with the nutrient content of feedstuffs and mixed diets. Chemical and physical parameters of wheat have been proposed as indicators of feed wheat quality. Traditionally, potential nutritional value differences in wheat can be expressed by means of physical parameters, such as density weight. However, in our study, density weight was not correlated with NE or AME, which is similar to the report by Zijlstra et al. [15], which showed that density was not correlated with DE content of wheat in 12 wheat samples but correlated with DE content in 12 wheat samples. It is apparent that density is a very weak contributor to predicting the nutritional value of wheat.

Using correlations and stepwise regression analysis, the equations used to predict NE content of wheat in ducks were established according to the significant linear relationship between NE, AME, NDF and ADF content. The results of the present study showed that the correlation coefficients between NE with EE, CF, NDF and ADF in this study were 0.432, −0.398, −0.835 and −0.676, respectively. Using stepwise regression to generate equations, NDF and ADF could be used as predictors to generate equations for predicting NE. They were consistent with most literature data showing the correlation for diets and individual ingredients of energy value with chemical composition and utilizing this high correlation by incorporating the different fiber fraction values into prediction equations, based otherwise on proximate analysis parameters. Zhao et al. [29] indicated that the AME, AMEn, TME, and TMEn of corn were negatively correlated with CF, ADF and NDF contents, in which NDF and GE were found to be useful for the ME prediction models in adult Pekin ducks. Wan et al. [30] suggested that NDF could be used as an effective indicator for the prediction of the TME value of wheat byproducts for ducks. The sample study reported by Noblet et al. [5] showed that ADF and NDF could be parameters for the prediction of NE content of diets for growing pigs. The other reason may be that NDF represents most of the fiber fractions of feedstuff, which were poorly digested by poultry such as ducks because of the lack of enzymes that can effectively degrade these complex carbohydrates [31]. There is hardly any doubt concerning the existence of a strong negative relationship between the amount of fiber in the diet and its nutritive value. Therefore, energy utilization efficiency of wheat is negatively and prominently affected by NDF content, which serves to make NDF as an effective predictor of NE of wheat for ducks. A significant improvement in the accuracy of prediction was obtained in Equation 2 (Table 6), when ADF, positively correlated with CF and NDF, was introduced into the prediction model. The results of the study also clearly showed that NE content of wheat could be more accurately predicted from the AME content with NDF and ADF, because the correlation coefficient between AME and NE is approximately 0.883 (*p* < 0.01). It was agreed that the accuracy of the NE equations based on both crude nutrient content and digestibility data was much higher than regression models that include either crude nutrient content or digestibility data [27].

## 5. Conclusions

In summary, wheat is one of the feed ingredients whose chemical composition is extremely variable. The NE content of wheat ranged from 6.81 to 9.12 MJ/kg in growing ducks. This experiment suggested that measuring the chemical composition of wheat samples may provide a quality parameter for NE content, especially for fiber. The accuracy of the NE prediction equation could be further improved when AME is taken into consideration.

## Figures and Tables

**Figure 1 animals-13-01097-f001:**
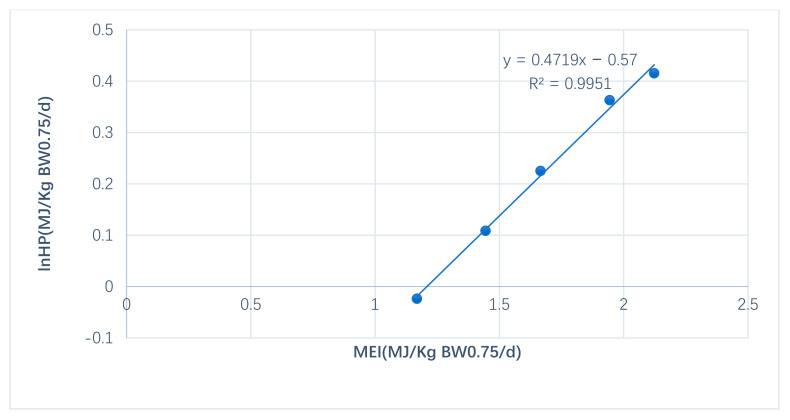
Determination of fasting heat production (FHP), lnHP (MJ/kg BW^0.75^d^−1^) as a function of MEI (MJ/kg BW^0.75^d^−1^) in duck. MEI, metabolizable energy intake; HP, heat production.

**Table 1 animals-13-01097-t001:** Composition and nutrient levels of the experimental diets (%).

Ingredients	Basis Diets	Test Diets	Nutrient Levels	Basis Diets	Test Diets
Corn	59.97	43.1	ME (MJ/kg)	12.11	12.18
Wheat	0	20	CP	20.50	20.58
Soybean meal	35.3	32.17	Ga	0.79	0.81
Soybean oil	1.2	1.2	P	0.69	0.70
Limestone	0.7	0.7	Lys	1.03	0.97
Dicalcicum phosphate	1.7	1.7	Met	0.46	0.46
Nacl	0.33	0.33	Trp	0.24	0.24
50%-Choline chloride	0.15	0.15	Thr	0.77	0.73
Vitamin premix ^1^	0.08	0.08	Cys	0.77	0.76
Mineral premix ^2^	0.25	0.25			
DL-Methionine	0.17	0.17			
L-Lysine.HCL	0.045	0.045			
Tryptophan	0.02	0.02			
Threonine	0.085	0.085			
Total	100	100			

^1,2^ Premix provided the following quantities of vitamins and micro-minerals per kilogram of complete diet for growing ducks: vitamin A, 4000 IU; vitamin B1, 2.0 mg; vitamin B2, 10 mg; vitamin D3, 2000 IU; vitamin E, 20 IU; vitamin K3, 2.0 mg; pantothenic acid, 20 mg; niacin, 50 mg; vitamin B6, 4.0 mg; vitamin B12, 0.02 mg; folacin, 1.0 mg; biotin, 0.15 mg; choline chloride, 1000 mg; Fe, 60 mg; Cu, 8.0 mg; Mn, 100 mg; Zn, 60 mg; Se, 0.30 mg; I, 0.40 mg.

**Table 2 animals-13-01097-t002:** Chemical and physical characteristic of 40 wheat samples (on dry matter) (%).

Characteristic	Mean	CV *	lowest	Highest
Chemical Composition				
Moisture	12.02	7.42	9.93	13.89
EE	1.90	12.90	1.31	3.29
CP	14.46	7.56	10.66	16.16
CF	3.07	8.77	2.65	3.73
Ash	1.76	17.55	1.31	3.29
NDF	9.56	11.54	6.80	12.15
ADF	2.58	15.54	1.69	3.92
Physical Characteristics				
Bulk weight g/L	775.30	4.95	660.4	863

* CV, coefficient of variation.

**Table 3 animals-13-01097-t003:** Energy concentration and NE/AME of 40 wheat samples.

Terms	Mean ± SD	CV %	Lowest	Highest
GE KJ/g	16.21 ± 0.19	1.16	15.71	16.54
AME MJ/kg	13.15 ± 0.73	5.67	11.03	14.34
NEm MJ/kg	2.91 ± 0.62	21.92	2.01	4.14
NEp MJ/kg	5.57 ± 1.20	22.14	3.49	7.36
NE MJ/kg	8.15 ± 0.52	6.40	6.81	9.12
NE/AME %	61.93 ± 1.86	3.01	57.92	65.12

GE, gross energy; AME, apparent metabolizable energy; NEm, net energy for maintenance; NEp, net energy for production; NE, net energy; NE/AME, the conversion rate of AME to NE; CV, coefficient of variation.

**Table 4 animals-13-01097-t004:** ME intake, retained energy and heat production for ducks under different feeding levels.

Feeding Levels	Ad Libitum Intake	Restricted Feeding 15%	Restricted Feeding 30%	Restricted Feeding 45%	Restricted Feeding 60%
MEI (MJ/kgBW^0.75^d^−1^)	2.10 ± 0.07 ^a^	1.94 ± 0.03 ^b^	1.67 ± 0.07 ^c^	1.44 ± 0.09 ^d^	1.17 ± 0.08 ^e^
RE (MJ/kgBW^0.75^d^−1^)	0.61 ± 0.04 ^a^	0.51 ± 0.03 ^b^	0.42 ± 0.02 ^c^	0.33 ± 0.01 ^d^	0.19 ± 0.03 ^e^
HP (MJ/kgBW^0.75^d^−1^)	1.51 ± 0.09 ^a^	1.43 ± 0.05 ^b^	1.25 ± 0.08 ^c^	1.11 ± 0.09 ^d^	0.98 ± 0.08 ^e^

MEI, metabolizable energy intake; RE, retained energy, heat production; In the same row, vaues with different letter superscripts mean significant difference (*p* < 0.05).

**Table 5 animals-13-01097-t005:** Pearson correlation coefficient between NE and AME, conventional composition of wheat samples.

	NE	AME	CP	CF	NDF	ADF	EE	Ash	Bulk Weight
NE	1	0.883 **	0.097	−0.398 *	−0.835 **	−0.676 **	0.432 **	−0.215	0.176
AME		1	0.090	−0.467 **	−0.757 **	−0.540 **	0.530 **	−0.169	0.199
CP			1	−0.025	−0.044	−0.178	0.314 *	−0.411 **	0.142
CF				1	0.425 **	0.403 **	−0.184	0.502 **	−0.575 **
NDF					1	0.569 **	−0.349 *	0.142	−0.173
ADF						1	0.063	−0.218	0.063
EE							1	−0.296	0.085
Ash								1	−0.677 **
Bulk weight									1

*, **, *p* < 0.05, *p* < 0.01, respectively. NE, net energy; AME, apparent metabolizable energy; CP, crude protein; CF, crude fiber; NDF, neutral detergent fiber; ADF, acid detergent fiber; EE, ether extract.

**Table 6 animals-13-01097-t006:** Equations of prediction of the NE (MJ/kg) values of wheat.

	Equations	R^2^	RSD	*p* Value
(1)	NE = 11.919 − 0.394 NDF	0.698	0.29	<0.001
(2)	NE = 12.151 − 0.315 NDF − 0.386 ADF	0.757	0.26	<0.001
(3)	NE = 0.629 AME − 0.120	0.780	0.25	<0.001
(4)	NE = 0.418 AME − 0.184 NDF + 4.408	0.845	0.21	<0.001
(5)	NE = 0.380 AME − 0.147 NDF − 0.274 ADF + 5.262	0.874	0.19	<0.001

NE, net energy; NDF, neutral detergent fiber; ADF, acid detergent fiber; AME, apparent metabolizable energy; RSD, residual standard deviation.

## Data Availability

Data is contained within the article. The data presented in this study are available in [Prediction of the Net Energy of Wheat from Chemical Analysis for Growing Ducks].

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
