# Peer review of "Prediction of the Net Energy of Wheat from Chemical Analysis for Growing Ducks"

_animals, 2023, doi:10.3390/ani13061097_

Round 1

Reviewer 1 Report

Overview and general recommendation:

Studies about the net energy content of ingredients are extremely relevant since energy is the most expensive component of diets. The net energy system has been widely studied in recent years in order to improve animal diets by reducing cost and increasing performance. Although this system is already widespread for pigs, the same does not occur in poultry. Several researchers have been supporting the implementation of a net energy system for poultry, however, more data about this topic is necessary, especially in duck nutrition.

The main idea of the present study is good, and the authors found relevant and coherent results. However, the manuscript needs to be revised and corrected. Additionally, the study would be more interesting if the authors had also determined the net energy value for each of the wheat varieties.

Specific Comments:

Abstracts:

Line 21: Please change the word “replicas” to “replicates”

Keywords: Please try to avoid using key words that have already been mentioned in the title.

Material and Methods:

Line 74: Forty different types of wheat were used? Please specify more if there were forty different types of wheat cultivars.

Line 80: Please change the word “fecal” to “excreta” since fecal refers only to the digestive content, disregarding the urine.

Line 80: I recommend to replace the word “method” to “technique”, since comparative slaughter is a technique supported by the energy balance method.

Line 81: Please change the word “method” to “analyses”.

Line 83: Avoid to use fractional days. I recommended to combine days and hours e.i., 5 days and 12 hours, or you can consider a full day, i.e., six days. Half a day will not influence the accuracy or precision of the experiment.

Line 87: “a total of 7 days” unnecessary fragment.

Line 91: Basal diet is more appropriate than basic diet.

Line 94: All numbers lower than eleven, should be write the number in full.

Line 95: Please detail how many days was consist the restricted feeding regimen (the feeding levels).

Line 95: Please change “excrement” to “excreta”.

Line 96: Please change “consistent” to “consist”.

Line 97: Please change the word “died” to “euthanized” and the word “neck dislocations” to “cervical dislocation”.

Line 102: Please, confirm the mathematical expression, because in the result is detail that was a LN expression, and in here was detail a LOG expression. Is important to be mentioned that the HP in function of MEI has an exponential behavior [HP=a*EXP(b*MEI)], the linear transformation involve the expression as LN(HP)=LN(a)+b*MEI or LOG(HP)=LOG(a)+b*LOG(MEI), both equation is right and can be express the behavior of the variables, but is important to detail what transformation was used on the M&M section and match with the result section.

Line 103: The NEm determination could be thought FHP measurement using indirect calorimetry method or estimated thought regression. For that, is important to mentioned that the FHP on this study was estimated and not calculated.

Line 104: I suggest to re-adequate this sub-topic (2.4), since the sub-topic 2.3. is used to inform about comparative slaughter and RE measurements. The difference between topics is about the aim of each trail. e.g., This can be renamed as: 2.3. NEm estimation through feeding levels. 2.4. RE determination by comparative slaughter technique for wheat cultivars.

Line 108: Please change the word “excrement” to “excreta”.

Line 112: “Finally, the RE values for each wheat diet were obtained for the meat ducks”. Repeated information, can be replaced by: The RE was expressed in the metabolic body weight bases.

Line 116: “Chinese meat duck feeding standard”. This is referred to the nutritional recommendation and management. If it is the nutritional recommendation, please provide the citation.

Line 124: I suggest to re-write and consider getting out some paragraphs, since that is repeated information in previous sub-topics.

Line 131: “logHP = a + b MEI (Lofgreen and Garrett, 1968)” It was before mentioned. Please, confirm if this is LN or LOG function.

Line 131 to 132: “Briefly, the HP and MEI were 131 measured and linear regression equation was generated between logHP and MEI, the FHP 132 (kJ/kg BW0.75d-1) was estimated by extrapolation when MEI was set at zero”. Please consider to remove this part, repeated information.

Line 144: “GE” Please detail the abbreviation meaning at first time that is mentioned on the main text.

Line 144: Please change the word “feces” to “excreta”. Only use feces if the urine was separated.

Line 147: The first authors that proposed the energy partitioning are Armsby and Fries. Overciting.

Line 170, Table 2: Please consider changing “fresh basis” to “as is base”.

Line 178: “LnHP” Please check if this is LN or LOG function.

Line 195: The correlation analyses between nutrients was not mentioned in the Material and Methods section. I suggest that you add this information.

Table 4: Should the predictors be correlated with AME?

Line 207: AME was used to predict ME? that paragraph is right? Please check this information.

Table 5, Eq. 5: I can see some limitations of this equation. The correlation that exists between NDF and ADF suggests that two both nutrients are related. Also, AME are correlated with NDF and ADF. Surprisingly, the stepwise procedures could be found a best solution with these nutrients. I strongly recommended to re-analyze the equations considering the correlation between nutrients and their statistical implication with the equation’s constructions. I am worried about that equation could be specific for their values, and the variance of ADF and NDF composition for the wheat’s types is not sufficient to evidence variations of NE from the variation of the other nutrients. Additionally, with your methodological procedure, can be calculated the wheat energy values (for each kind of wheat). Could be interesting for the readers to know the NE value for each wheat.

Discussion:

Line 241: “NSP” is the abbreviation to non-starch polysaccharides? Please detail the abbreviation meaning at first time that is mentioned on the main text.

Line 243: Please change the word “energy absorbed” to “energy intake”.

Line 250: “fasting method” which method? Calorimetry?

Line 253: By activity you mean physical activity? Please clarify.

Lines 254 to 257: Interesting sentence, when the authors say values below and above maintenance, which maintenance would that be?

Line 260: “CS” What CS means? Please detail the abbreviation meaning at first time that is mentioned on the main text.

Line 260 to 261: This sentence should be in the results part. Repeated information.

Lines 262 to 264: I do not see how information about the FHP of quails can contribute to the discussion of your data, since the body mass of quails is lower than ducks. I strongly suggest that you search for papers by authors who have worked with net energy, either for broilers and laying hens, e.i.: Barzegar, Jaap van Milgen, Sakomura, Shubiao Wu.

Line 273: I suggest to change the word “In conclusion” to “In summary”.

Line 276: Effective energy is a concept described by Emmans, it should not to be used unless you are going to discuss about it.

Lines 280, 283 and 285: Please change the word “conversion efficiency” to “energy use efficiency” or “energy efficiency”.

Reviewer 2 Report

The present study was undertaken to determine the NE content of wheat using comparative slaughter in growing ducks and establish NE prediction equation based on the chemical composition of wheat, to provide experimental basis for the construction and application of NE system in meat duck poultry. The research is topical and may be considered for publication after moderate revision,

1, There is no experimental design associated or indicated in the study and this is a major flaw and must be indicated in the study to improve the quality of the study.

2, Please indicate the statistical model and the regresion model used in the study based on the study design and indicate what each of the variables represent in the model.

3. Please indicate the method used for the mean separation in the study, This is missing and is very crucial.

Reviewer 3 Report

The manuscript deals with the prediction of the net energy of wheat from chemical analysis for growing ducks. The evaluation of net energy contents independence of the species is an ongoing topic regarding a sustainable animal nutrition. Hence the manuscript fits in the scope of the journal. However, there are some minor remarks which should be considered before the manuscript can be approved for publication.

Line 39-40: This sentence can be removed from the manuscript

Point 2.1.: please describe in detail how the bulk intensity was evaluated

Line 84-84: please describe in detail how homogenization of slaughtered animals was carried out. This is very important for a correct measurement of the gross energy values.

Line 99: Abbreviation is explained later (line 100).

Point 2.5. The authors are writing in a wrong time. They should switch back to the past.

Line 132: replace production by performance

Table 4: include Pearson correlation (if this was the case) in the headline of table 4

Point 4.1.: please describe clearly if the values refer to dry matter basis

Line 287-288: Absolutely bring other poultry NE values for wheat and discard the comparison with pigs.

Round 2

Reviewer 1 Report

To the Authors:

The correlation between nutrients is very important to be verified and studied carefully. It is regrettable that such relevant research data has been lost. I strongly suggest that future studies use an online platform for data storage.